# Synthesis and Characterization of N and Fe-Doped TiO_2_ Nanoparticles for 2,4-Dimethylaniline Mineralization

**DOI:** 10.3390/nano12152538

**Published:** 2022-07-24

**Authors:** Emerson Faustino, Thalita Ferreira da Silva, Rebeca Fabbro Cunha, Diego Roberto Vieira Guelfi, Priscila Sabioni Cavalheri, Silvio César de Oliveira, Anderson Rodrigues Lima Caires, Gleison Antonio Casagrande, Rodrigo Pereira Cavalcante, Amilcar Machulek Junior

**Affiliations:** 1Institute of Chemistry, Federal University of Mato Grosso do Sul (UFMS), Av. Senador Filinto Muller, 1555, CP 549, Campo Grande 79074-460, MS, Brazil; emerson.faustino@ifro.edu.br (E.F.); thalita.quim@gmail.com (T.F.d.S.); rebecafabbro@hotmail.com (R.F.C.); diegoguelfi@outlook.com (D.R.V.G.); priscilasabioni@hotmail.com (P.S.C.); scolive@gmail.com (S.C.d.O.); gleisoncasag@gmail.com (G.A.C.); 2Federal Institute of Education, Science and Technology of Rondônia, Rodovia RO-257, s/n—Zona Rural, Ariquemes 76870-000, RO, Brazil; 3Department of Sanitary and Environmental Engineering, Dom Bosco Catholic University, Campo Grande 79117-900, MS, Brazil; 4Optics and Photonics Group, Institute of Physics, Federal University of Mato Grosso do Sul (UFMS), Campo Grande 79070-900, MS, Brazil; anderson.caires@ufms.br; 5School of Technology, University of Campinas—UNICAMP, Paschoal Marmo, 1888, Limeira 13484-332, SP, Brazil

**Keywords:** 2,4-dimethylaniline, N-Fe-doped TiO_2_, characterization

## Abstract

The present study aimed to evaluate the feasibility of developing low-cost N- and Fe-doped TiO_2_ photocatalysts for investigating the mineralization of 2,4-dimethylaniline (2,4-DMA). With a single anatase phase, the photocatalysts showed high thermal stability with mass losses of less than 2%. The predominant oxidative state is Ti^4+^, but there is presence of Ti^3+^ associated with oxygen vacancies. In materials with N, doping was interstitial in the NH_3_/NH^4+^ form and for doping with Fe, there was a presence of Fe-Ti bonds (indicating substitutional occupations). With an improved band gap energy from 3.16 eV to 2.82 eV the photoactivity of the photocatalysts was validated with an 18 W UVA lamp (340–415 nm) with a flux of 8.23 × 10^−6^ Einstein s^−1^. With a size of only 14.45 nm and a surface area of 84.73 m^2^ g^−1^, the photocatalyst doped with 0.0125% Fe mineralized 92% of the 2,4-DMA in just 180 min. While the 3% N photocatalyst with 12.27 nm had similar performance at only 360 min. Factors such as high surface area, mesoporous structure and improved E_bg_, and absence of Fe peak in XPS analysis indicate that doping with 0.0125% Fe caused a modification in TiO_2_ structure.

## 1. Introduction

2,4-Dimethylaniline (2,4-DMA, also called 2,4-Xylidine, C.A.S. No. 95-68-1) is an aromatic amine used as a precursor for some textile dyes and veterinary drugs [1,2]. Considered a persistent contaminant, 2,4-DMA has already been detected in industrial effluents and groundwater. For example, 2,4-DMA was detected in textile industry wastewater even after anaerobic treatment processes [3]. Toxicologically, 2,4-DMA has been classified as a group 3 carcinogen and has adverse effects on the aquatic ecosystem [4]. Furthermore, 2,4-DMA is a model compound used in degradation studies because its oxidation produces intermediates that facilitate mechanistic studies [5,6].

Persistent contaminants exhibit chemical and biological resistance to mineralization through traditional water treatment methods. Therefore, the search for innovative treatment technologies has increased. Advanced oxidation processes (AOPs) are an alternative to completely mineralizing for numerous contaminants, as it is not a selective process [7]. AOPs are able to remove various types of pollutants from attacks by reactive oxygen species (ROS) generated in the system [8,9,10,11,12,13,14,15,16]. Heterogeneous photocatalysis (PC) are AOPs that involves the activation of a photocatalyst by light irradiation. Under specific conditions, photogenerated charge carriers (e^−^_CB_ and h^+^_VB_) undergo interfacial transfer and promote redox reactions with adsorbed molecular oxygen, water molecules, and hydroxide ions, producing various types of ROS such as superoxide, hydrogen peroxide, and hydroxyl radicals (HO^●^) [13,17]. Many semiconductors, such as TiO_2_ [18], WO_3_ [14], ZnO [19], and BiVO_4_ [20], have been used to photodegrade emerging contaminants. TiO_2_ is the most investigated photocatalyst due to its low toxicity, low cost, chemical stability over a wide pH range, thermal stability, minimizing e^−^_CB_/h^+^_VB_ pair recombination, and the band gap energy (E_bg_) being able to oxidize water molecules generating HO^●^ radicals. The principle of PC involves the activation of a semiconductor by solar or artificial irradiation, Equation (1). Photons absorbed with energy greater than the E_bg_ promote an electron from the valence band to the conduction band with the simultaneous generation of a hole (h^+^_VB_) in the valence band. The adsorbed water (H_2_O_ads._) is oxidized in the h^+^_VB_ generating HO^●^, which can subsequently oxidize the organic contaminant (R), Equations (2) and (3). Or the R can be oxidized directly in the h^+^_VB_, Equation (4).
TiO_2_ + hυ → TiO_2_ (e^−^_CB_ + h^+^_VB_)(1)
h^+^_VB_ + TiO_2_-H_2_O_ads._ → HO^●^ + H^+^(2)
HO^●^ + R → CO_2_ + H_2_O(3)
h^+^_VB_ + R → CO_2_ + H_2_O(4)

TiO_2_ has a high E_bg_ (3.2 eV for anatase) and is excited only by ultraviolet light irradiation (λ < 385 nm) [21]. The photocatalytic activity of TiO_2_ can be improved by doping or modifying its crystalline structure in order to increase the optical absorption of radiation in the visible range and decrease the rate of recombination of photogenerated charges [22]. Light absorption of TiO_2_ in the visible region can be improved via metal- [23,24,25] or non-metal-doping [18,26,27]. Several approaches for the synthesis of TiO_2_ have been reported, such as sol-gel, hydrothermal, solvothermal, and spray pyrolysis [28,29,30,31]. Among them, the sol-gel method has been considered a good solution to produce nanomaterials, presenting advantages such as homogeneity, ease and flexibility in introducing doping at high concentrations, simple synthesis process, and providing nanomaterials with high purity. During sol-gel preparation, a TiO_2_ sol or gel is formed by precipitation through the hydrolysis and condensation of a titanium alkoxide [18]. Jadhav et al. synthesized N-doped TiO_2_ doped with 25% N using urea as a precursor. The material synthesized by them degraded 70% of the methylene blue in 180 min of the experiment under visible light [32]. Synthesis using ammonium salt as precursor has also been reported and its photocatalytic activity showed a 90% removal in 40 min of rhodamine B [33]. Yalçın et al. synthesized Fe-doped photocatalysts and tested the photoactivity on 4-nitrophenol degradation degrading 80% in 120 min [34].

In this work, we synthesized and characterized N-, Fe-, and N-Fe-doped TiO_2_ nanoparticles via sol-gel. Furthermore, we evaluated the photocatalytic activities of TiO_2_ NPs through 2,4-DMA mineralization. This research provides new results on the influence of nitrogen and iron on TiO_2_ on 2,4-DMA mineralization. It is worth mentioning that, to the best of our knowledge, no studies were reported involving the photocatalysis process with N-, Fe- doped TiO_2_ for the mineralization of 2,4-DMA, with good results of mineralization rates in 6 h in an unprecedented manner.

## 2. Materials and Methods

### 2.1. Synthesis of Photocatalysts

The synthesis of photocatalysts via sol-gel followed the procedure described below [18,35]. A volume of 19.10 mL of titanium IV isopropoxide (97%, Sigma Aldrich, St. Louis, MO, USA) and 16.10 mL of glacial acetic acid (99.7%, Alphatec, Carlsbad, CA, USA) was added to a beaker and left under magnetic stirring. After homogenization, 19.10 mL of 2-propanol (99,6%, Merck, Kenilworth, NJ, USA) was added and left under stirring for 60 min. Thereupon, 30 mL of deionized water (resistivity > 18 MΩ cm obtained from Gehaka DG500 UF system, São Paulo, SP, Brazil) acidified with 1 mL of nitric acid (65.0%, Vetec, Duque de Caxias, RJ, Brazil) was added dropwise to the above solution. At the end of the addition, the solution was stirred for another 120 min. Finally, the resulting product was maintained at 40 °C for about 48 h for gel formation. The gel was dried in an oven at 100 °C for 24 h. The product has been macerated and placed in a muffle furnace at 450 °C for 4 h.

The synthesis of TiO_2_ doped with N, Fe, and N-Fe followed the same procedure described above, but there was the addition of ammonium hydroxide (25%, Merck, Kenilworth, NJ, USA) as a source of N and iron III nitrate (98%, Vetec) as a precursor of Fe. N-doped TiO_2_ were prepared in the different weight content of nitrogen (%): 1.0, 2.0, 3.0, 4.0, 5.0, 6.0, 7.0, and 9.0 (*w*/*w*). Fe-doped TiO_2_ were prepared in the different weight content of iron (%): 0.0125, 0.025, 0.05, 0.1, 0.25, and 1.0 (*w*/*w*). N-Fe co-doped TiO_2_ were prepared by fixing 0.025% Fe and varying only N by 2, 3, and 4% (*w*/*w*). These concentrations of N and Fe were chosen based on previous work [18,35]. The effects of the presence of N and Fe on the physical and chemical properties of photocatalyst particles were investigated by increasing or decreasing the photocatalytic response in 2,4-DMA mineralization. Figure 1 presents a schematic representation showing the steps of the synthesis.

### 2.2. Photocatalysts Characterization

The nanoparticles synthesized from pure TiO_2_, N, and Fe-doped TiO_2_ are characterized by thermogravimetric (TG) and differential thermogravimetric (DTG) curves recorded by the TA Instruments thermal analyzer-TGA Q50. Approximately 10 mg of each sample was used and the thermocurves are recorded in the range of room temperature to 850 °C in atmosphere of nitrogen (N_2_) gas. The crystalline structure of the synthesized pure TiO_2_, N, and Fe-doped TiO_2_ nanoparticles is carried out by powder X-ray diffraction (XRD) technique employing LabX XRD-6100, Shimadzu. The XRD analysis is done in the 2θ range of 10° and 70° using Co Kα radiation of wavelength 17,889 Å. Fourier transform infrared spectroscopy (FTIR) analysis of the samples was performed with a PerkinElmer FTIR spectrophotometer, Frontier model. The samples were prepared by the KBr pellet pressing method, at room temperature and in the region of 4000–450 cm^−1^. The morphological of the materials was studied by Scanning Electron Microscopy (SEM) in a JSM-6380LV JEOL microscope and Transmission Electron Microscopy (TEM) in a model JEM 2100 LaB_6_ JEOL microscope. Using the ImageJ software and TEM images, the diameter of the particles in the different samples was measured [18]. Diffuse UV-visible reflectance spectroscopy (DRS) analyzes were performed on the samples using a PerkinElmer Lambda 650 UV/Vis/NIR spectrophotometer operating in the range of 200–800 nm. From the DRS studies, a graph of the modified Kubelka-Munk function [F(R) × hν]^1/2^ versus absorbed light energy was constructed for indirect band gap energy (E_bg_) calculations. The surface area and pore size distribution of the samples were calculated by the Brunauer-Emmett-Teller (BET) method using the ASAP 2010, Micromeristic surface area analyzer. X-ray photoelectron spectroscopy (XPS) measurements were performed on a EA125 Sphera spectrometer. The elemental composition of the samples was determined by X-ray photoelectron spectroscopy (XPS). The analyzes were performed on an Omicron, model EA125 Sphera equipped with a hemispherical electron analyzer. All measurements were performed in an ultra-high vacuum (UHV) chamber with pressure between 10^−8^ and 10^−12^ mbar. The C1s peak of carbon contamination at 284.60 eV was used as a binding energy reference.

### 2.3. Photocatalytic Mineralization

The photocatalytic activity of the synthesized materials was tested for the mineralization of 2,4-DMA. In a typical procedure, 350 mL (0.1 mmol L^−1^) of 2,4-DMA solution and 175 mg of photocatalyst were transferred to a glass reactor. The suspension was kept under constant stirring in the dark for 30 min, ensuring the adsorption-desorption equilibrium of the test molecule. A UVA lamp (18 W, DULUX L-OSRAM, λ = 340–415 nm and 8.23 × 10^−6^ Einstein s^−1^) was used as irradiation. After the adsorption-desorption equilibrium was established, the lamp was turned on, and 4 mL aliquots were collected at different times. The mineralization was followed by the total organic carbon (TOC) analysis using a Shimadzu TOC VCPN analyzer. The calibration accuracy values with LOQ = 0.180 mg L^−1^ and LOD = 0.053 mg L^−1^ were obtained. The reuse experiments lasted 180 min and various aliquots were collected and analyzed by TOC. The photocatalyst used was collected by vacuum filtration, washed with deionized water (resistivity > 18 MΩ cm obtained from Gehaka DG500 UF system, São Paulo, Brazil) and dried in an oven at 80 °C (Medicate, MD 1.2). The reuse experiment cycle was repeated three times.

Apparent pseudo-first order kinetic constants (*k*_2,4-DMA_) (Equation (5)) to 2,4-DMA were calculated according to the following:(5)$$k_{2,4-\mathrm{DMA}}=\frac{ln\frac{{\left[\mathrm{TOC}\right]}_{0}}{{\left[\mathrm{TOC}\right]}_{t}}}{t}\;\;\;\;$$

## 3. Results and Discussion

### 3.1. Characterization of N-, Fe-, and N-Fe Doped TiO_2_ Photocatalysts

Appendix A (see Appendix A) shows the results of the TG and DTG analysis for the synthesized N-, Fe-, and N-Fe doped TiO_2_ nanoparticles. It can be seen in Appendix A, that the synthesized materials have a high thermal stability independent of doping. The mass losses were less than 3%, which is consistent with previous works [25,36,37]. In the DTG curves (Appendix A), an endothermic event before 100 °C can be seen and is associated with evaporation of adsorbed water and/or decomposition of residual organic matter.

Some characteristic peaks of TiO_2_-based photocatalysts can be seen in Figure 2A. The bands at 700 cm^−1^ are typical of the Ti-O and Ti-O-Ti bonds present in TiO_2_. The bands at 1650 cm^−1^ and 3200 cm^−1^ are associated with the O-H vibrational modes of adsorbed water. The N-doped photocatalysts present bands close to 1500 cm^−1^ and 3200 cm^−1^, referring to the vibration of the N-H group [32]. The vibrational bands of the Fe-O bond are in the range of 400 to 700 cm^−1^, but they were not observed due to the low concentration of iron used.

Figure 2B shows XRD patterns for the investigated photocatalysts. The anatase phase peaks at 29.54° (101) and 44.14° (200) compared to ICDD Card no. 00–021–1272, are present in the samples. The peaks of the rutile phase are absent in the diffractograms. The calcination temperature (450 °C) does not promote the formation of this crystalline phase [38]. The other peaks at 56.36° (105), 63.50° (116), 64.96° (200), and 74.44 (2015) are also due to the anatase phase belonging to the tetragonal crystal system [13]. From the Rietveld refinement (see Appendix A) the special group I4_1_/amd (141) and a density of 3.90 g cm^−3^ for TiO_2_, 0.125%Fe-TiO_2_ and 3%N-TiO_2_. The presence of the anatase phase in the photocatalysts is consistent with the literature [25,39].

Obtaining the anatase phase is satisfactory, as it allows a more effective absorption of photons than the rutile phase of TiO_2_. The anatase phase of TiO_2_ increases the photocatalytic performance compared to the rutile phase as it increases the redox power of the charge carriers and can enhance the adsorb of hydroxyl groups on its surface [40]. In the case of Fe doping, the absence of peaks at 31.07° corresponding to Fe_3_O_4_ and 36.18° and 41.30° corresponding to FeTiO_3_ in Figure 2B indicates that Fe^3+^ cations did not react with TiO_2_ [34,41]. The absence of these oxides is desirable because their presence reduces photocatalytic activity [34]. Therefore, substitutional doping of Fe^3+^ occurred in the photocatalysts instead of interstitial doping.

The morphology and size of N-, Fe-, and N-Fe doped TiO_2_ photocatalysts were investigated by SEM (Figure 3) and TEM (Figure 4). According to Figure 3A–G, the morphology of the synthesized oxides does not follow a defined pattern, presenting clusters in a slightly rounded shape on their surface. In some cases, the clusters are larger than others, as seen in the A (pure), D (3% N), E (9% N), and F (2% N + 0.025% Fe) images in Figure 3.

Figure 4 shows the TEM micrographs and the particle diameter distribution histogram. The data show that the TiO_2_ photocatalysts have an average particle diameter of 16.16 nm (Figure 4A), while 0.0125% Fe (Figure 4B) and 3% N (Figure 4C) have 14.45 nm and 12.27 nm, respectively. TiO_2_ doping resulted in particle reduction, regardless of the dopant used.

N_2_ adsorption-desorption isotherm at liquid nitrogen temperature was conducted to analyze the pore size and surface area of N-, Fe-, and N-Fe doped TiO_2_ photocatalysts, as shown in Figure 5. BET surface areas (*S*_BET_), pore volume (*V*_p_) and pore diameters (*D*_p_) are shown in Table 1. The samples show surface areas of 78.67, 84.73, and 95.70 m^2^ g^−1^ for TiO_2_, 0.0125% Fe, and 3% N, respectively. In Figure 4, we clearly see that the samples are agglomerates formed by monodisperse primary particles, with sizes of 16.16, 14.45, and 12.27 nm for TiO_2_, 0.0125% Fe, and 3% N, respectively. Table 1 indicates that the nanoparticles are porous with mean pore diameters of 9.7 to 9.0 nm and narrow pore size distributions (Figure 5). Farhangi et al. reported that Fe doping could decrease TiO_2_ crystallization and also slightly restrict TiO_2_ crystallite growth [42]. The presence of N- and Fe- caused a decrease in the nanoparticle and an increase in its surface area. Large surface area and mesoporous structure are favorable characteristics for obtaining high photocatalytic activity.

The chemical composition and electronic states of the photocatalyst constituents: pure TiO_2_, 0.0125% Fe and 3% N were investigated by XPS, as shown in Figure 6. According to the survey spectrum Ti, O, and C peaks are evident. For all samples can observed the presence of peaks related to Ti 3p, Ti 3s, C 1s, Ti 2p, O 1s, and Ti 2s at around 26.0, 39.0, 286.0, 462.0, 533.0, and 566.0 eV, respectively. The origin of carbon is common in XPS analyses, resulting from the strong adsorption of a contaminating layer or hydrocarbon from the carbon strip of the instrument [35,43]. The chemical composition of photocatalysts is shown in Table 2. As seen in Figure 7A–C, two peaks at approximately 458.5 and 464.3 eV that correspond to typical Ti^4+^ 2p_3/2_ and Ti^4+^ 2p_1/2_, are observed for all samples [34,44]. In Figure 7A–C, the Ti 2p_3/2_ peak at 457.1 eV and the Ti 2p_1/2_ peak at 462.8 eV correspond to the Ti^3+^ species [45]. The O 1s signals for all samples (Figure 7D–F) can be divided into two peaks at 529.8 and 531.0 eV, corresponding to Ti-O and surface −OH bonds, respectively [44]. For comparison purposes, the contribution of the peaks at 531.0 eV at 0.0125% Fe (24.9%) and 3%N (17.1%) is lower than that of TiO_2_ (42.0%), indicating fewer oxygen vacancies in the doped catalysts. Oxygen vacancies on the surface of the particle tend to promote more −OH bonds [46]. Metallic Fe signals were not observed at 710.7 eV for 2p_3/2_ and 724.3 eV for 2p_1/2_, which indicates the absence of Fe_2_O_3_ [34]. As the presence of metallic Fe was not detected, the iron cations probably occupied substitutional positions due to the similar radius of Fe^3+^ (0.64 Å) and Ti^4+^ (0.68 Å), forming a solid Fe-Ti solution during the sintering process (<1 at Fe%) [47]. The peak at 399.1 eV in Appendix A refers to the N 1s that are present in the 3% N sample, this peak is characteristic of interstitial N. Values above 400 eV can be assigned to groups NO_2_^-^, N_2_ and NH_x_. As shown by the results obtained from FTIR (Figure 2A) there are vibrations in the N-H group, so it can be said that the peak of N 1s is nitrogen in the NH_3_/NH_4_^+^ form. It is considered that nitrogen does not replace the oxygen atom in the crystal matrix, it was incorporated to form Ti–O–N bonds on the surface of the material [25,48].

The UV-visible absorption spectra of N-, Fe-, and N-Fe doped TiO_2_ photocatalysts are presented in Figure 8. The absorption spectra in Figure 8A shows intense absorption in the region of 300 to 450 nm, resulting from the intrinsic band gap of TiO_2_. The 1% Fe photocatalyst showed improved absorption in the visible range (in the 400 to 700 nm) when compared to other doped photocatalysts. The E_bg_ (Figure 8B) of the samples was estimated using Tauc’s formula and their values are found in Table 3. Pure TiO_2_ had an E_bg_ of 3.16 eV, a value close to that reported in the literature [18]. The E_bg_ values for N-doped catalysts showed a small difference in relation to pure TiO_2_. The likely reason behind the decrease in the E_bg_ is that the nitrogen inside the TiO_2_ doped material replaces the Ti ions inside the material [32], as seen in Figure 8. For the Fe-doped photocatalysts, a gradual decrease is observed with increasing Fe concentration. This effect may be associated with the new energy levels introduced in the TiO_2_ band gap caused by Fe^3+^ and Fe^2+^ ions. The 1% Fe photocatalyst presented an E_bg_ of 2.82 eV. This value is 0.34 eV lower than the band gap of pure TiO_2_ anatase nanoparticles (3.16 eV), which can be caused by Ti^3+^ sites (vacancies of oxygen) [49] as a role of the intermediate trap state formed in the calcination process. Barkhade and Banerjee also detected a decrease in E_bg_ from 3.4 eV for pure TiO_2_ to 2.5 eV for a TiO_2_ sample doped with 1.0% Fe (mol %). They attributed the decrease to the insertion of Fe^3+^ ions into the TiO_2_ lattice in which conduction band overlap occurred due to the Ti (d-orbital) of TiO_2_ and the metal (d-orbital) orbital of Fe^3+^ ions. Furthermore, the doping of Fe^3+^ ions induces the formation of new electronic states (Fe^4+^ and Fe^2+^) that span across the TiO_2_ band gap [50].

The photocatalytic efficiency of a catalyst depends on some aspects such as light absorption, charge separation and types of catalysts. The fundamental reaction in photocatalysis occurs with the absorption of light [51], which leads to charge separation and formation of active species, as shown in Equations (1)–(4). The charge carriers generated can separate and then recombine. But they can also migrate to the surface of the catalyst and recombine in surface traps or undergo interfacial electron transfer with the reduced species. The recombination kinetics is related to the presence of impurities in the bulk and recombination centers on the catalyst surface. Therefore, the quantum yield of a doped catalyst is often lower than that of the related undoped material. However, light capture plays an important role in determining the rate of degradation of organic molecules. Consequently, here the improved light-gathering capacity of N-, Fe- and N-Fe-doped TiO_2_ photocatalysts would be beneficial to increase their photocatalytic activity.

### 3.2. Photocatalytic Activity

The photocatalytic activity of N-, Fe-, and N-Fe doped TiO_2_ was measured using a 2,4-DMA solution under UVA light. The effects of nitrogen content (1, 2, 3, 4, 5, 6, 7, 9%), iron (0.0125, 0.025, 0.05, 0.1, 0.25, and 1%) and N-Fe co-doping (3% N +0.025% Fe, and 4% N + 0.025% Fe) were investigated by TOC (Total Organic Carbon) with their respective pseudo-first-order constants (*k*_2,4-DMA_).

Figure 9A shows the relationship between 2,4-DMA mineralization and irradiation time for each x% N photocatalyst during PC, and in Figure 9B their respective *k*_2,4-DMA_. In 360 min of the experiment, the 3% N and 4% N photocatalysts mineralized 100% and presented a *k*_2,4-DMA_ of 0.007 min^−1^ and 0.006 min^−1^, respectively. Furthermore, the values of *k*_2,4-DMA_ were 1.8 and 1.5 times higher than mineralization with pure TiO_2_ (Table 4). Analyzing the experiment time at 240 min, we see that increasing the percentage of nitrogen up to 3% (*w*/*w*) promotes an increase in mineralization.

The results also revealed that the increase above 3% of N did not improve the mineralization performance, presenting similar results to those observed for pure TiO_2_. The higher mineralization rate for the 3% N photocatalyst is the result of the increase in surface area (78.66 m^2^ g^−1^) in relation to pure TiO_2_ (95.69 m^2^ g^−1^). It was also observed that the pore diameter was reduced from 9.72 nm (pure TiO_2_) to 9.03 nm (3% N), and the mean particle diameter obtained by the MET results was reduced from 16.16 nm to 12.27 nm for 3% N. With the improved contact surface, more ROS can be generated thus increasing photocatalytic activity [18,35].

Figure 9C illustrates the effect of iron content on x% Fe photocatalytic activity within 360 min reaction time. It is observed that as the doping content increases, the photocatalytic activity decreases. This decrease in photoactivity is associated with the fact that Fe^3+^ traps both e^−^_CB_ and h^+^_VB_ up to a certain amount; as the doping content exceeds this ideal amount, it acts as a recombination center for the photogenerated charge carriers decreasing photocatalytic activity [47,52]. Thus, the 0.0125% Fe and 0.025% Fe photocatalysts mineralized 100% of 2,4-DMA in just 240 min. Furthermore, the values of *k*_2,4-DMA_ 0.010 min^−1^ for 0.0125% Fe and 0.025% Fe (Figure 9D, Table 4) were 2.6 times higher than mineralization with pure TiO_2_. In the presence of low levels of Fe doping, the charge carriers are well separated, thus increasing the efficiency of the photocatalyst. The stability of the 0.0125%Fe photocatalyst was performed and in three successive cycles, there was a reduction of only 6%. This reduction indicates good stability of the studied photocatalyst. Fe^3+^ can serve as photogenerated hole chaser (Equation (1)). Fe^4+^ reacts with hydroxyl ions adsorbed on the surface to produce HO^●^ radicals (Equations (6) and (7)). Alternatively, Fe^4+^ can also react with photogenerated electrons, improving photocatalytic activity (Equation (8)). The trapped holes would oxidize hydroxyls adsorbed on the surface to produce hydroxyl radicals [47,53].
Fe^3+^ + h^+^_VB_ → Fe^4+^(6)
Fe^4+^ + OH^−^(ads.) → Fe^3+^ + HO^●^(7)
Fe^4+^ + e^−^_CB_ → Fe^3+^(8)

Photocatalysts co-doped with N- and Fe- were synthesized by fixing 0.025% Fe and varying by 3 and 4% N, such amounts of N- and Fe- were selected from the photocatalytic activities reported here. Interestingly, the junction of N- and Fe- did not show good results, as can be seen in Appendix A (see Appendix A). With co-doping between N- Fe- only 50% of the 2,4-DMA was mineralized and the *k*_2,4-DMA_ value was two times lower than that of pure TiO_2_ (Table 4 and Appendix A). The increase in the surface area of the co-doped photocatalysts may have occurred, which contributed to the lack of the desired synergistic effect and to improve the photocatalytic activity. Yuan et al. reported the synthesis of N-Fe co-doped TiO_2_ photocatalysts and tested their photocatalytic activity on 2-methylisoborneol degradation. Using a xenon lamp the highest removal efficiency was achieved when the catalyst was co-doped with 0.001% Fe and 0.5% N, with a 2-methylisoborneol removal efficiency of 99.78% in 240 min. They observed that concentrations higher than 5% of N decrease the photoactivity due to an increase in the surface area of the catalysts [54].

Table 5 shows the mineralization results of N- Fe-doped TiO_2_ obtained in the present investigation and results of previously published studies [51,52,53,54,55,56]. For a better comparison, some parameters such as photocatalyst type, experimental conditions, model molecule for PC, light source conditions, and mineralization rate were also summarized in Table 5. In the literature, the mineralization rates found are achieved using a low volume of solution and a large amount of catalyst.

## 4. Conclusions

This study provided an understanding between N- and Fe-doped photocatalysts in the mineralization of 2,4-DMA. The main results found were that:Nanoparticles from 16.16 to 12.27 nm were synthesized by the sol-gel method;The crystalline phase was exclusively anatase with a density of 3.90 g cm^−3^;The decrease in surface area from 97.70 to 78.67 m^2^ g^−1^ was observed to have mesopores;Doping with Fe at the investigated concentrations promoted substitutional positions, increasing the formation of Fe-Ti bonds. For nitrogen, there was the presence of NH_3_/NH^4+^ species indicating interstitial doping;The 2,4-DMA mineralization was viable with high removal rates.

Therefore, our findings offer the opportunity to reconsider studies taking into account the mineralization of emerging pollutants. This avoids the use of expensive chromatographic techniques and toxicity studies as mineralization converts the pollutant into CO_2_, H_2_O, and inorganic anions.

## Figures and Tables

**Figure 1 nanomaterials-12-02538-f001:**
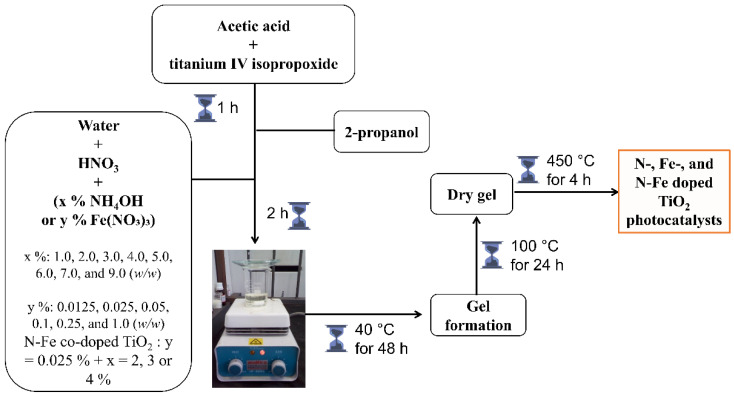
Representative scheme showing the steps of the catalyst synthesis process.

**Figure 2 nanomaterials-12-02538-f002:**
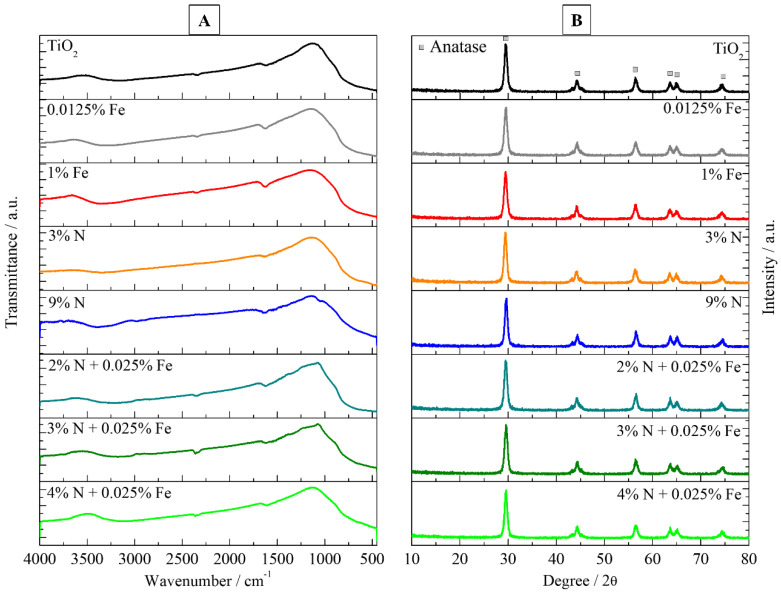
(**A**) FTIR spectra, and (**B**) X-ray diffraction patterns of N-, Fe-, and N-Fe doped TiO_2_ photocatalysts.

**Figure 3 nanomaterials-12-02538-f003:**
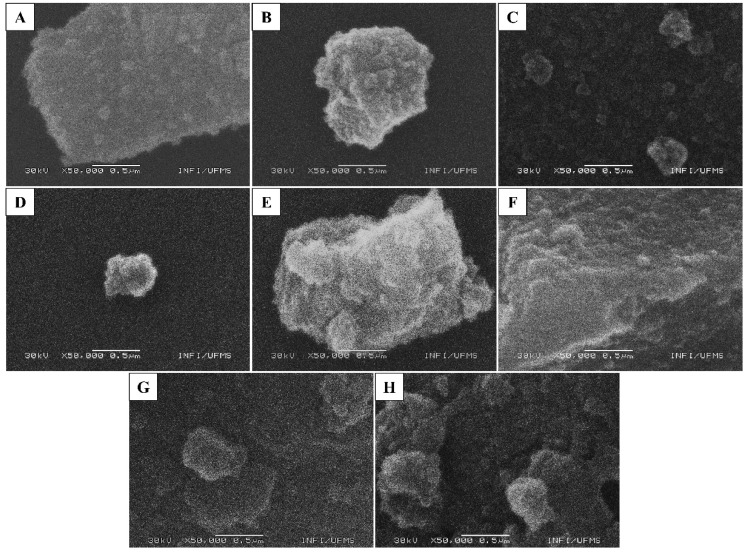
SEM image of the surfaces of N-, Fe-, and N-Fe doped TiO_2_ photocatalysts. (**A**) TiO_2_ pure, (**B**) 0.0125% Fe, (**C**) 1% Fe, (**D**) 3% N, (**E**) 9% N, (**F**) 2% N + 0.025% Fe, (**G**) 3% N + 0.025% Fe, and (**H**) 4% N + 0.025% Fe.

**Figure 4 nanomaterials-12-02538-f004:**
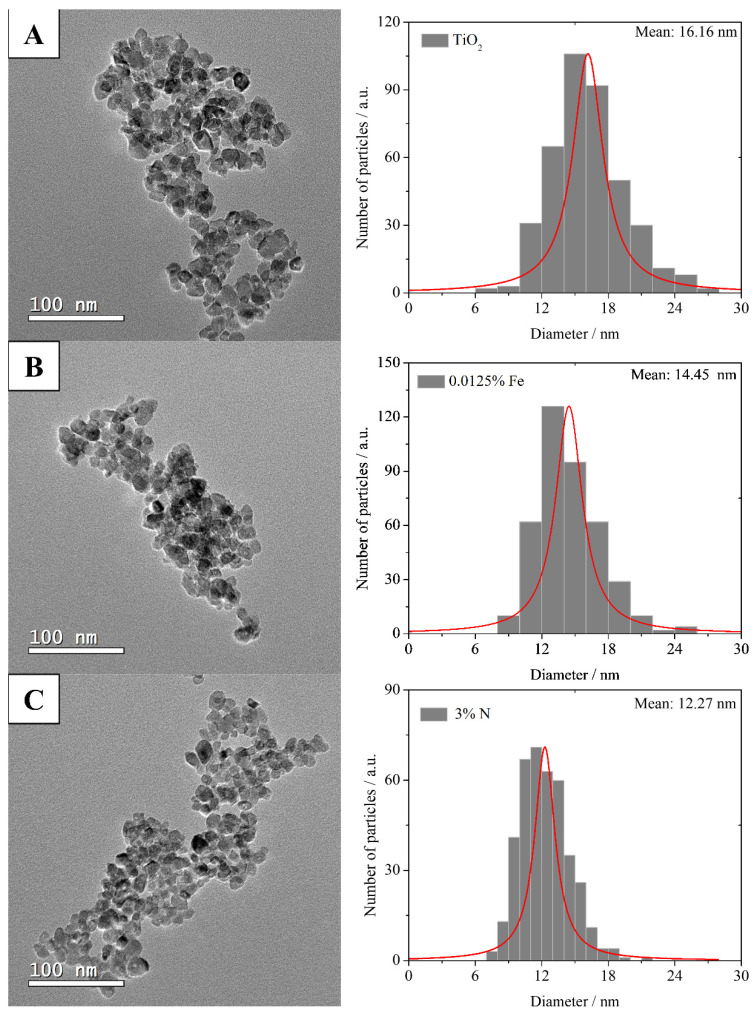
TEM images and their size distributions of particles (**A**) TiO_2_, (**B**) 0.0125% Fe, and (**C**) 3% N.

**Figure 5 nanomaterials-12-02538-f005:**
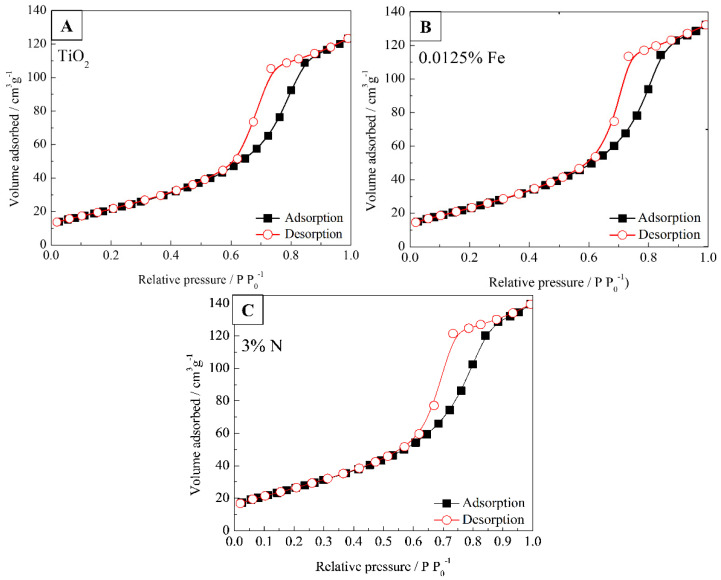
Nitrogen adsorption-desorption isotherms of (**A**) TiO_2_, (**B**) 0.0125% Fe, and (**C**) 3%N.

**Figure 6 nanomaterials-12-02538-f006:**
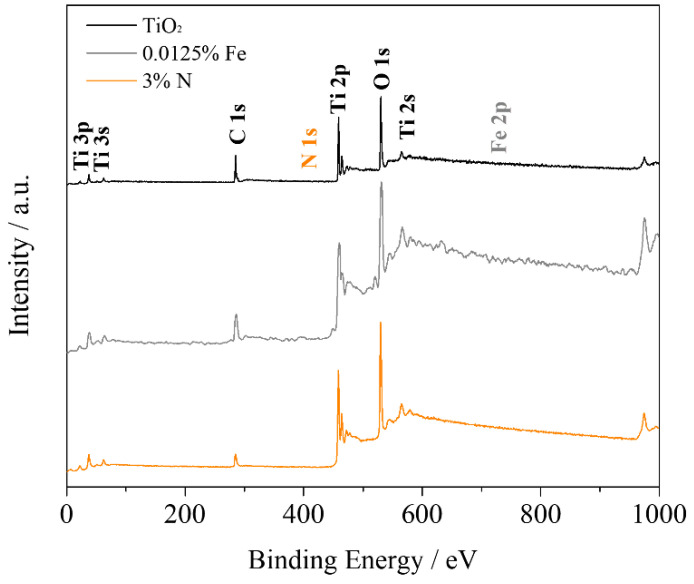
XPS spectra for the TiO_2_, 0.0125% Fe, and 3%N samples.

**Figure 7 nanomaterials-12-02538-f007:**
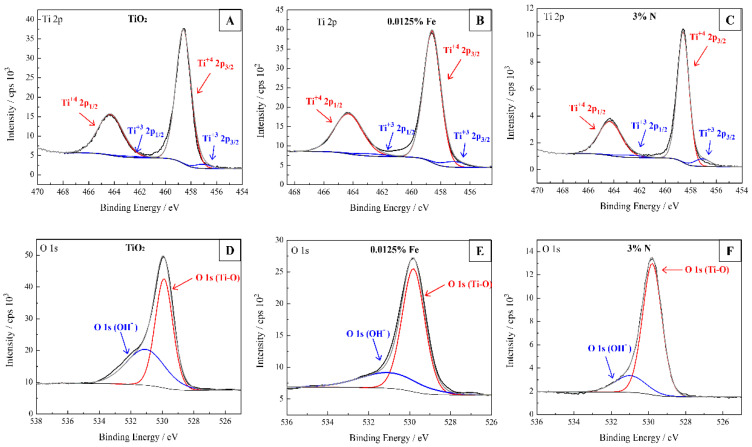
High-resolution XPS Ti 2p and O 1s spectra of TiO_2_ (**A**,**D**), 0.0125% Fe (**B**,**E**) and 3% N (**C**,**F**).

**Figure 8 nanomaterials-12-02538-f008:**
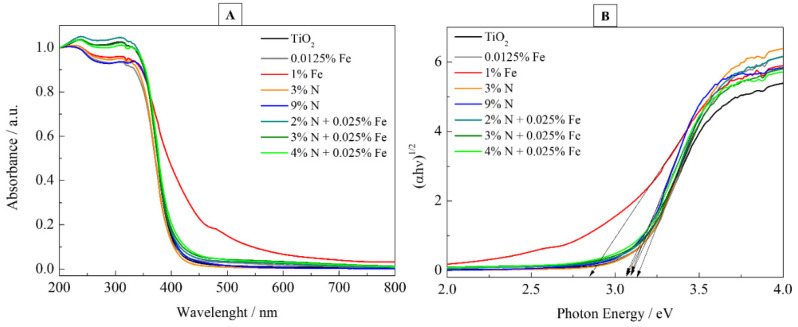
(**A**) UV-Vis absorbance spectra for the synthesized photocatalysts. (**B**) Tauc plots of photocatalysts with different % N and Fe.

**Figure 9 nanomaterials-12-02538-f009:**
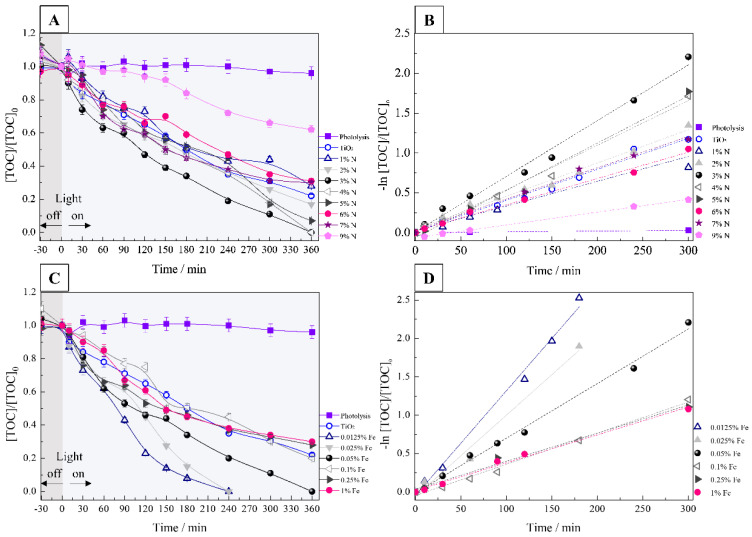
(**A**,**C**) Normalized 2,4-DMA mineralization curves obtained from the PC treatment of 350 mL of a solution of 0.1 mmol L^−1^ of 2,4-DMA with 175 mg of photocatalyst, irradiated by a UVA lamp of 18 W (**B**,**D**) pseudo-first-order kinetics for PC treatment.

**Table 1 nanomaterials-12-02538-t001:** BET surface areas (*S*_BET_), pore volume (*V*_p_), and pore diameters (*D*_p_) for the N-, Fe-, and N-Fe doped TiO_2_ photocatalysts.

Photocatalysts	*S*_BET_ (m^2^ g^−1^)	*D*_P_ (nm)	*V*_P_ (cm^3^ g^−1^)
TiO_2_	78.67 ± 1.50	9.73	0.19
TiO_2_ + 0.0125%Fe	84.73 ± 1.47	9.68	0.20
TiO_2_ + 3%N	95.70 ± 1.38	9.04	0.22

**Table 2 nanomaterials-12-02538-t002:** Binding energies/eV and contribution of the high-resolution spectra of Ti 2p present in the photocatalysts. Values are obtained from high-resolution XPS spectra in Figure 6.

Photocatalysts	Composition Ti	Binding Energies/eV Ti 2p_1/2_	Binding Energies/eV Ti 2p_3/2_	Contribution/ %
TiO_2_	Ti^4+^	464.30	458.58	96.84
	Ti^3+^	462.80	457.08	03.16
0.0125% Fe	Ti^4+^	464.30	458.58	94.20
	Ti^3+^	462.80	457.08	05.80
3% N	Ti^4+^	464.31	458.59	97.64
	Ti^3+^	462.81	457.09	07.38

**Table 3 nanomaterials-12-02538-t003:** Band gap energy values for all photocatalysts.

Photocatalysts	E_bg_/eV
TiO_2_	3.16
TiO_2_ + 0.0125%Fe	3.08
TiO_2_ + 1%Fe	2.82
TiO_2_ + 3%N	3.14
TiO_2_ + 9%N	3.07
3% N + 0.025% Fe	3.05
4% N + 0.025% Fe	3.06

**Table 4 nanomaterials-12-02538-t004:** Removal percentage mineralization and pseudo-first-order constant.

Photocatalysts	% Dopant (*w*/*w*)	Mineralization Rate (%)		Pseudo-First-Order Constant	
		in 240 min	in 360 min	*k* _2,4-DMA_	R^2^
Photolysis	-	0	4	0.0001	0.961
TiO_2_	0	65	78	0.004	0.983
TiO_2_ + N	1	57	72	0.003	0.993
	2	64	83	0.004	0.988
	3	81	100	0.007	0.989
	4	64	100	0.006	0.989
	5	55	93	0.006	0.983
	6	53	69	0.003	0.991
	7	62	70	0.004	0.990
	9	28	38	0.002	0.999
TiO_2_ + Fe	0.0125	100	100	0.010	0.992
	0.025	100	100	0.010	0.980
	0.05	80	100	0.010	0.994
	0.1	55	80	0.004	0.990
	0.25	63	72	0.004	0.983
	1	62	70	0.004	0.990
TiO_2_ + %N + 0.025 Fe	3	31	39	0.002	0.997
	4	48	52	0.002	0.992

**Table 5 nanomaterials-12-02538-t005:** Comparison of N-TiO_2_ and Fe-TiO_2_ photocatalysts with others.

Photocatalyst	Experimental Conditions				Model Molecule	Mineralization	Ref
	Mass/mg	Volume/mL	Light Source	Analysis Time/h			
0.0125% Fe TiO_2_	175	350	UV-A	4	12 ppm 2,4-DMA	100%	Present study
2 mol% Fe-TiO_2_	50	100	100 W fluorescent bulb	1	10 ppm Indigo Carmine	68%	[55]
N-TiO_2_	1	5	UV-A/solar/UV-vis	8	Microcystin-LR	80%	[56]
gC_3_N_4_	30	30	Visible-light irradiation	3	20 ppm 2,4-dichlorophenol	15.2%	[57]
BiPO_4_ + H_2_O_2_	50	100	11 W low pressure lamp (λ = 254 nm)	2	phenol	40%	[58]
pg-C_3_N_4_/Co_3_O_4_/CoS	5	50	500 W xenon lamp	2	30 ppm Bisphenol F	51.9%	[59]
Pt_SA_/g-C_3_N_4_	25	100	Xenon light	4	10 ppm p-chlorophenol	70%	[60]

## Data Availability

Not applicable.

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
