# Peer review of "Synthesis and Characterization of N and Fe-Doped TiO2 Nanoparticles for 2,4-Dimethylaniline Mineralization"

_nanomaterials, 2022, doi:10.3390/nano12152538_

Round 1
Reviewer 1 Report
Authors addressed all the review comments in the revised manuscript.
Author Response
The authors are grateful for the contributions made by reviewer 1. Thank you very much!
Reviewer 2 Report
The manuscript titled “Synthesis and characterization of N and Fe-doped TiO2 nanoparticles for 2,4-dimethylaniline mineralization” presented by Emerson Faustino , Thalita Ferreira Da Silva , Rebeca Fabbro Cunha , Diego Roberto Vieira Guelfi , Priscila Sabioni Cavalheri , Silvio César De Oliveira , Anderson Rodrigues Lima Caires , Gleison Antonio Casagrande , Rodrigo Pereira Cavalcante * , Amilcar Machulek Junio. The manuscript presented after resubmitting. I have noted that authors did a good work under the manuscript and answered/commented some questions. However, there is one major issue concerning to XPS analysis of the samples:
- I should constated that the authors did not answer to questions about the XPS. Firstly, the nature of XPS spectrum of 2p core-level is that the Ti 2p has a Ti2p3/2-Ti2p1/2 doublet with spin-orbit splitting and the ratio between peaks is 2:1! In the other word, if one introduces in fitting model Ti2p3/2 peak one should introduce the Ti2p1/2 peak at higher energy with 2 times lower integral intensity. And vice versa, of course. What we see at Figures 7a-7c the Ti2p3/2 core-level has two peaks corresponded (as authors considered) to Ti4+ and Ti3+, but the Ti2p1/2 has only one peak corresponded to Ti4+ doublet structure. Why the Ti2p1/2 peak corresponded to Ti3+ doublet structure is absent? A new insight to physics? It is absolutely inaccessible. Moreover, it looks that there is do not needed to add the additional doublet to fit the Ti2p spectrum – one doublet (corresponded to Ti4+ state) is sufficient to fit the spectrum. Additionally, the binding energy of Ti2p3/2 peak corresponded to Ti3+ state is lower the one corresponded to Ti4+ state. For example, for Ti4+ state the binding energy of Ti2p3/2 peak is about 459.0 eV when for Ti3+ state – about 457.5 eV (see, for example, http://www.xpsfitting.com/2008/09/titanium.html).
- The authors discuss the presence of peak in the noise (Fig. S3). Please. Provide the N1s spectrum with high S/N ratio. The high-resolution spectrum could not be measured with 0.25 eV step.
Thus, the authors did not answer major questions about XPS. The current version could not be accepted for publication - authors should revise the XPS section.
Author Response
Dear Reviewer,
Comments about XPS have been answered to the best of our ability. We fundamentally agree with all comments made by reviewer #2 and have incorporated the corresponding revisions into our manuscript. Our detailed responses to Reviewer #2's comments are provided in the attached file below.

Reviewer 3 Report
Revision made by the author satisfactory and the present form of manuscript should be accepted for publication.
Author Response
The authors are grateful for the contributions made by reviewer 3. Thank you very much!
Round 2
Reviewer 2 Report
Authors have redone the XPS section and finalized the manuscript. The manuscript could be accepted for publication.
This manuscript is a resubmission of an earlier submission. The following is a list of the peer review reports and author responses from that submission.
Round 1
Reviewer 1 Report
Dear Editor: I would like to express my deep thanks for inviting me to review the manuscript ID: nanomaterials-1725491
Title: Synthesis and characterization of N and Fe-doped TiO2 nano-particles for 2,4-dimethylaniline mineralization
Authors: Emerson Faustino, Thalita Ferreira da Silva, Rebeca Fabbro Cunha, Diego Roberto Vieira Guelfi, Silvio César de Oliveira, Anderson Rodrigues Lima Caires, Gleison Antonio Casagrande, Rodrigo Pereira Cavalcante, Amil-car Machulek Junior
Comments:
Introduction:
Need to rewrite introduction part. For example, importance of this work, explain the synthesis process the objectives and novelty and so on. Follow below references and cited them
1. M. Zeshan, I. A. Bhatti, M. Mohsin, M. Iqbal, N. Amjed, N. AlMasoud,T.S. Aloma “Remediation of pesticides using TiO2 based photocatalytic strategies: A review” Chemosphere 300 (2022) 134525
2. B.T. Lee, J.K. Han, A.K. Gain, K.H. Lee, F. Saito, “TEM microstructure characterization of nano TiO2 coated on nano ZrO2 powders and their photocatalytic activity” Materials Letters 60 (17-18), (2006) 2101-2104
3. M.A.E. Wafi, M.A. Ahmed, H. S. Abdel-Samad, H.A.A Medien, “Exceptional removal of methylene blue and p-aminophenol dye over novel TiO2/RGO nanocomposites by tandem adsorption-photocatalytic processes” Materials Science for Energy Technologies 5 (2022) 217-231
Materials and Methods
Explain in detail the characterization section
Results and discussion:
In Figure. 2 please identify all peak in XRD profile for example TiO2 different phases.
Please provide EDS analysis data in In Figure 3 and identify the doped phases.
Conclusion part:
Please rewrite the conclusion part in bullet points.
RECOMMENDATION
After reviewing the enclosed manuscript for “Nanomaterials”, the present manuscript contains some kinds of scientific analysis but it is mandatory required to modify according to the preceding remarks. So, the manuscript can be publication after major revision.
Reviewer 2 Report
Manuscript deals with the Synthesis and characterization of N and Fe-doped TiO2 nano- 1 particles for 2,4-dimethylaniline mineralization. But publication point of view some major modifications necessary.
1) There are few grammatical mistakes. Please check the manuscript for grammar and English.
2) Literature survey is very poor in introduction part, to enrich it add some references related with this work.
i)Journal of Colloid and Interface Science 606, (2022) 454-463, ii) Journal of Colloid and Interface Science 582, (2021) 1058-1066, iii) Surfaces and Interfaces 24, (2021) 101075
3) What is novelty of the present work? Rewrite it at the end of introduction section.
4) Add stability study of photocatalyst material.
5) In fig. 9 add error bars.
6) Compare your results of photocatalytic degradation experiments with other researcher work in tabular form.
7) How doping affects the catalytic performanace.
Reviewer 3 Report
The manuscript titled “Synthesis and characterization of N and Fe-doped TiO2 nanoparticles for 2,4-dimethylaniline mineralization” presented by Emerson Faustino, Thalita Ferreira da Silva, Rebeca Fabbro Cunha, Diego Roberto Vieira Guelfi, Silvio César de Oliveira, Anderson Rodrigues Lima Caires, Gleison Antonio Casagrande, Rodrigo Pereira Cavalcante, Amilcar Machulek Junior deals with the TiO2 based photocatalysts for mineralization of 2,4-dimethylaniline. However, there are issues that could be clarified or took into account:
- Line 22: TiO3+ – what the authors want to say, Ti5+ state?
- The abstract badly written. For example, “The 0.0125% Fe doped photocatalyst with 14.45 nm and improved surface area to 84.73 m2g-1 mineralized to 92% at 2,4-DMA in just 180 min”. It could not be understood. The abstract should be re-written to more understandable form.
- An introduction does not allow reader to evaluate the actuality of study.
- Line 114: X-ray diffraction (w/o analysis)
- Line 116: IFTR?
- Line 163: XRD diffractograms = X-ray diffraction diffractograms (tautology). It should be corrected to XRD patterns.
- Fig. 2b all peaks should be marked and corresponded to phases.
- Line 178-180: “The formation of these metal oxides is desirable, as their photocatalytic activity is poor and they also occupy active sites on the TiO2 surface, resulting in a decrease in the number of HO● radicals” The sentence has contradictory parts.
- Fig. 3 is low quality figure! What is the scale of images?
- The XPS data description is low quality. 1. The authors found the presence of Ti4+ cations with corresponded binding energies 459.3 eV (Ti2p3/2 peak) and 464.9 eV (Ti2p3/2 peak) and it is expected for TiO2. And, it is strange, the authors found peak at 459.8 eV that they corresponded to Ti2p1/2 of Ti3+ cations. The question is where the first component of Ti2p3/2-Ti2p1/2 doublet? According to spin-orbit splitting and basic physics the Ti2p3/2 peak should be found at approx. 454.2 eV. But such binding energy of Ti2p3/2 peak is corresponded to Ti in the metal state. Moreover, no peaks at 454.2 eV with high intensity are observed (I should stress that the intensity of 2p3/2 component is two times higher then one of 2p1/2 component). 2. The idea that the absence of peaks in the Fe2p region is due to “forming a solid Fe-Ti solution during the sintering process” is very doubt. In my opinion the authors did not collect the Fe2p spectra enough time. 3. The authors refer to Fig. S3 (N1s spectrum) where they found peak in the noise. No comments.
The choice of methods to study the catalysts looks irrationally. TGA, FTIR and SEM did not give any information that could clarify anything – all catalysts are similar. XPS analysis was poorly done, thus, the authors made incorrect conclusions. The XPS analysis should be done for all catalysts – the surface concentration of cations should be presented. The structure of manuscript and data analysis and presentation should be redone. The manuscript demands a strong proofreading (typos, English correction). The current version of manuscript demands many corrections and additional studies and could not considered for publication.